# Impact of Groundwater Abstraction on Hydrological Responses during Extreme Drought Periods in the Boryeong Dam Catchment, Korea

**Sanghyun Park [1], Hyeonjun Kim [1,2,\*] and Choelhee Jang [2]**

[1] Department of Civil and Environmental Engineering, University of Science & Technology, 217 Gajeong-ro, Yuseong-gu, Daejeon 34113, Korea; sanghyun0385@kict.re.kr

[2] Department of Land, Water and Environment Research, Korea Institute of Civil Engineering & Building Technology, 283 Goyang-daero, Ilsanseo-gu, Goyang-si 10223, Korea; chjang@kict.re.kr

[\*] Correspondence: hjkim@kict.re.kr; Tel.: +82-31-910-0003

**Abstract:** Groundwater withdrawal results in a significant depletion of groundwater storage due to the frequency and intensity of droughts and increasing irrigation demands. To ensure the sustainable use of groundwater resources, it is necessary to accurately simulate the groundwater behavior of catchments using a surface–groundwater integrated runoff model. Most of the existing catchment runoff models have been applied to surface water management; thus, integrated runoff analysis studies that consider the interaction between surface water and groundwater are required. Due to the intensive agricultural sector in Korea and the position of rice as the staple in the Korean diet, more than 50% of groundwater abstraction is used for irrigation. Therefore, it is very important to understand the hydrological interrelationships between agricultural areas and the entire watershed. This study aimed to compare and analyze the groundwater levels in the mountainous areas and paddy field areas in the Boryeong Dam catchment through a surface–groundwater integrated runoff simulation using the Catchment Hydrologic Cycle Assessment Tool model, and to compare the hydrological responses in wet years (2010–2012) and dry years (2014–2016). The maximum difference in the monthly groundwater level in the dry years compared to the wet years was 1.07 m at the forest catchment and 0.37 m at the paddy catchment. These results indicate that the impact of drought on the groundwater level of paddy catchments is not significant compared to the forest catchments; however, drought slows the recovery of the groundwater level before the rainy season, thereby limiting the agricultural groundwater use in the catchment.

**Keywords:** irrigation; drought; runoff simulation; groundwater; CAT; Boryeong Dam

## 1. Introduction

Climate change has been shown to be a principal factor changing the hydrological cycle on a global scale [1]. The drought cycle pattern in South Korea has been shortened recently, and droughts of varying intensity have occurred every year since 2013 [2]. The Boryeong Dam, which serves as a major water source in the middle-western region of Korea, recorded its lowest storage volume in 2015 due to the continuous shortage of precipitation since 2014. The dam levels were stabilized by limiting domestic water consumption in the region and constructing a water supply tunnel of 21.9 km, connecting the Geum River to the Boryeong Dam catchment [3]. The Ministry of Agriculture, Food and Rural Affairs of Korea [4] has classified the drought warning steps according to the storage of the dam into four stages: the concern stage (less than 70% storage in a normal year), warning stage (less than 60%), caution stage (less than 50%), and serious stage (less than 40%). The water storage volume of the Boryeong Dam in 2015 decreased to 18.87%, reaching the serious stage due to the extreme drought that occurred in the central region of Korea [5]. The drought in the central region continued, and the water storage rate reached 8.29% in 2017—the

lowest water storage level since the construction of the Boryeong Dam [5]. Jung et al. [6] analyzed the drought index using unstructured data of the climate, streamflow, and rainfall of Boryeong city in the dry years of 2014–2016. Yu et al. [7] analyzed the drought transitions in meteorological and agricultural droughts, including the 2015 drought, and Kim et al. [8] assessed the water shortage of the Boryeong Dam by applying future climate change scenarios.

As the frequency and intensity of droughts have increased, the importance of groundwater as a sustainable water resource has been recognized [9]. According to the Third Comprehensive Long-Term Water Resources Plan (2016) of the Ministry of Land, Infrastructure, and Transport of Korea, the groundwater level is decreasing every year due to groundwater consumption and urbanization. Groundwater use, which was 2.57 billion $m^3$ in 1994, increased by ~60% to 4.1 billion $m^3$ in 2014, indicating a very high dependence on groundwater use in Korea [10]. Groundwater is an important water resource that dominates streamflow, especially in the dry season, and its variability directly contributes to the health of a stream [11]. Brunner et al. [12] analyzed MODFLOW simulations related to surface water–groundwater interactions. Mukherjee et al. [13] showed that the depletion of streams is related to the reduction in baseflow caused by groundwater storage depletion in adjacent aquifers.

Groundwater storage is not only affected by meteorological factors but can also be significantly affected by groundwater extraction for agricultural purposes [14], and groundwater withdrawals for agricultural purposes have a significant impact on the groundwater system of the watershed itself [15]. In addition, the paddy field area accounts for approximately 8.21% of the Korean territory [16], and about 51% of the total amount of groundwater consumption is used only for agriculture [17]. Therefore, the impact of groundwater withdrawals for agricultural purposes on the hydrological cycle of the catchment must be closely examined to accurately estimate the impact of drought on groundwater levels in paddy field areas in Korea. Since the aquifer is dynamically connected to the nearby sub-catchments, it is necessary to quantitatively analyze not only the groundwater movement in paddy fields but also the hydrological interactive responses between the paddy field areas and the whole watershed. Leng et al. [18] performed and analyzed Community Land Model 4.0 (CLM4) simulations driven by downscaled/bias-corrected historical simulations and future projections from general circulation models (GCMs) to investigate the effects of irrigation on global water resources. Ferguson and Maxwell [19] evaluated and compared the impacts of large-scale climate change and local-scale water management practices in an agricultural watershed using a fully integrated model of groundwater, surface water, and land surface processes. Sorooshian et al. [20] investigated the impacts of agricultural irrigation on hydrological processes in California using a regional climate model (RCM), an offline land surface model output, and available in situ observations and remote sensing data. Kannan [21] developed a modeling approach for canal irrigation systems and irrigation best management practices (BMPs) to simulate the hydrological balance of irrigated watersheds based on the water requirements of crops, the number and frequency of irrigation systems, and critical crop water requirement stages. Ozdogan et al. [22] described an original technique for applying satellite-derived irrigation data within a land surface model (LSM) and evaluated the effects on irrigation in the United States. Essaid et al. [23] examined changes in streamflow, groundwater discharge to the stream, and stream temperature resulting from irrigation practices using a watershed-scale surface water and groundwater flow integrated model.

The Boryeong Dam catchment is representative of most Korean catchments, with a steep and mountainous catchment including paddy field areas. Even though this catchment is not a regional-scale watershed, with an area of only 163 $km^2$, it has complex interactive hydrological cycle processes due to human activities such as groundwater abstraction, agricultural reservoirs, and export and import water supply systems in the catchment. However, most studies on hydrological runoff modeling in Korea have focused on the

surface runoff analysis and have not suggested human activities including groundwater pumping for irrigation demand and water intake systems in the watershed.

In this study, a runoff simulation of the Boryeong Dam catchment was conducted using a physical parameter-based and surface–groundwater integrated hydrological model for paddy fields in Korea, including a paddy module that considers the ponding depth and levee height of the irrigation area of the watershed. The groundwater levels of the paddy and forest areas of the watershed were analyzed and compared with two consecutive periods of wet years (2010–2013) and drought years (2014–2016).

## 2. Materials and Methods

### 2.1. Study Area Description

The study area is the Boryeong Dam catchment located in Chungnam Province (the middle-western region of South Korea) (Figure 1), and the area of the catchment is 163 km$^2$, with an average slope of 40.19% and an average elevation of 232.56 m. The average annual rainfall of the watershed is 1.244 mm, and the average monthly temperature of the catchment during the study period was −4.5 to 27.3 °C, indicating that the seasonal temperature fluctuation was very large. The Boryeong Dam, located in the Ungcheon stream of the Geum River, is a multi-purpose dam that was constructed in 2000. The Boryeong Dam has a water storage capacity of 116.9 million m$^3$ and supplies seven adjacent cities and counties [24].

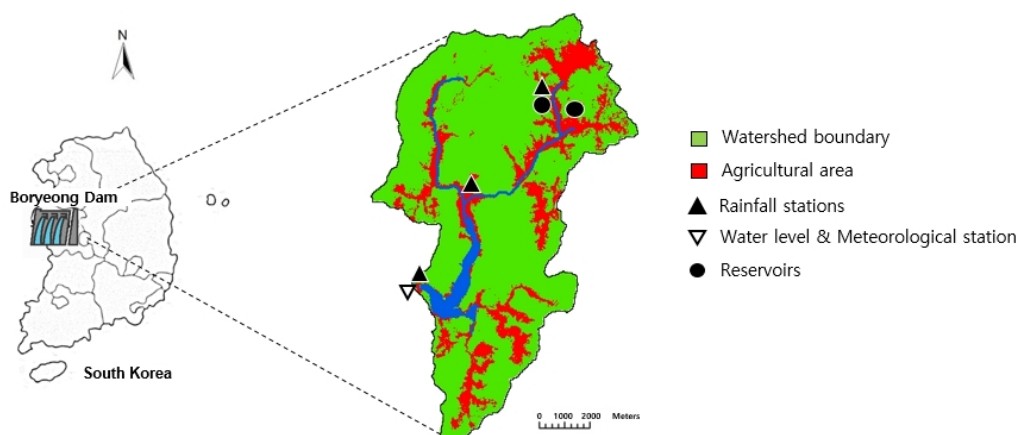

**Figure 1.** Location of the Boryeong Dam catchment including the rainfall stations, reservoirs, water level, and meteorological station.

### 2.2. Data Collection

The runoff simulation was conducted for the years 2000 to 2019, and 20 years of daily rainfall, streamflow, and meteorological data were collected to establish the Catchment Hydrologic Cycle Assessment Tool (CAT) model. Rainfall data were collected from the Boryeong Dam station, Dohwadam station, and Samsan station operated by the Korea Water Resources Corporation (K-water) [5], and the meteorological data used for estimating the evapotranspiration using the Penman–Monteith method, such as minimum and maximum temperature (°C), humidity (%), sunshine hours, and wind speed (m/s), were collected from the Boryeong Meteorological Station operated by the Korea Meteorological Administration [25].

Soil parameters required for model simulation were extracted from a 1:25,000 scale soil map of the Rural Development Administration of Korea [26], and groundwater aquifer parameters were extracted from the Boryeong Region Groundwater Research Report 2007 [5,10]. The land cover map of the Ministry of Environment [27] was used for classification into forest, urban, and paddy fields, and the input data for the model simulation were derived by dividing the watershed into pervious and impervious areas.

### 2.3. CAT Model

The CAT is a physical parameter-based distributed hydrological model that allows the quantitative evaluation of the long- and short-term water cycle of catchments [28]. It is a link–node connecting model designed to simulate runoff, including infiltration, evapotranspiration, and groundwater flow for each spatial unit, and it divides the hydrological cycle process into pervious and impervious areas. The major physical parameters required for the initial simulation are slope, soil type, land use, aquifer, and river information. The concept and development of the basic model were based on the unconfined aquifer and single soil layer assumptions. The model categorizes the incoming rainfall that falls on pervious, impervious, and paddy field areas, providing the surface flow, infiltration, or evapotranspiration (Figure 2).

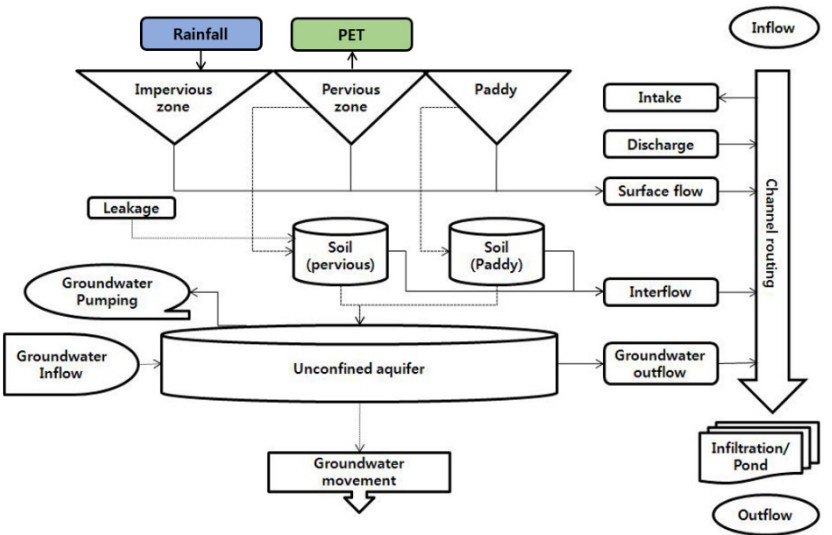

**Figure 2.** Schematic diagram of the CAT model (CAT 3.0 user's manual, 2017).

The CAT model has been applied in various studies for different catchments. Jang et al. [29,30] assessed the long-term hydrological behavior of agricultural reservoirs in the Idong catchment and analyzed the impact of future climate change on the hydrological responses and runoff in the Gyeongan–Cheon River Basin. Birhanu et al. [31] assessed the results of five hydrological models, including the CAT, applied in 10 catchments in Korea. Choi et al. [32] analyzed a short-term CAT runoff simulation and carried out a sensitivity analysis of soil parameters using three infiltration methods. Lee and Cho [33] analyzed the hydrological cycle of four catchments in Ulsan city, Korea, using the CAT, and Miller et al. [34] investigated the changes in stormwater runoff caused by the transformation of rural landscapes into peri-urban areas in the UK using the CAT.

To simulate the runoff process in the paddy field (Figure 3), the soil and groundwater layers were divided in the same way as the pervious area of the watershed. Artificial drainage facilities can be included in the soil layer to accommodate underground culvert drainage and pipe drainage; however, only surface drainage by levee height was considered in this study according to the field situation of the catchment. Surface drainage occurs when the ponding depth is greater than the height of the surface drain water threshold. The surface drain equations are

$$Q_s = \alpha\sqrt{H_s - H_p}\,(H_s > H_p) \tag{1}$$

$$Q_s = 0 \ (H_s \leq H_p) \tag{2}$$

where $Q_s$ is the discharge from the surface (m$^3$/s), $\alpha$ is the drainage coefficient of the surface drain levee in the paddy (mm$^{0.5}$/h), $H_s$ is the ponding depth of the paddy (m), and $H_p$ is the height of the surface drain levee of the paddy (m).

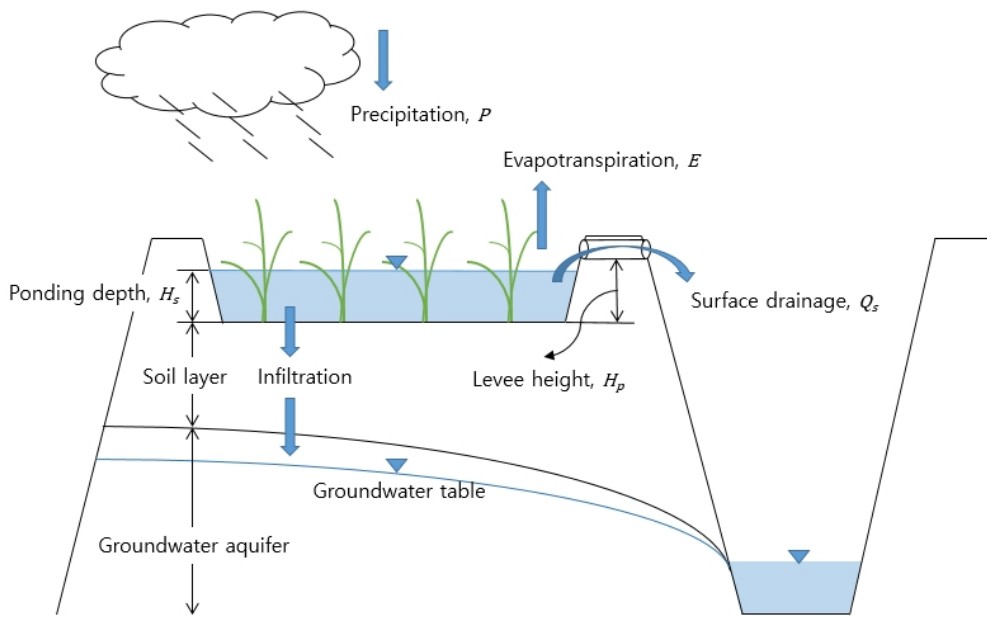

**Figure 3.** Hydrological cycle in the paddy field in the CAT model.

The hydrological interrelation between surface water and groundwater was calculated according to Darcy's law, with flows based on the hydraulic conductivities, river stage, and groundwater level. The interrelation between the river level and the groundwater level was calculated using Equations (3) and (4). Equation (4) was applied when the river level was higher than the groundwater level in the vicinity; otherwise, Equation (3) was applied.

$$Q_r = K_{sr} A_r \tag{3}$$

$$Q_r = K_{sr}\left(\frac{h - H_r}{b_r}\right) A_r \tag{4}$$

$$Q_g = K_{sr} \frac{\partial h}{\partial x} \cdot l \cdot T \tag{5}$$

$$Q_{in} - Q_{out} = A \cdot S \frac{dh}{dt} \tag{6}$$

where $Q_r$ is the inflow into the river or recharge from the river (m$^3$/s), $Q_g$ is the groundwater flow (m$^3$/s), $Q_{in}$ and $Q_{out}$ are the inflow and outflow of the aquifer (m$^3$/s), respectively, $K_{sr}$ is the saturated hydraulic conductivity of the riverbed (m/s), $A_r$ is the area of the riverbed (m$^2$), $b_r$ is the riverbed thickness (m), $h$ is the groundwater level (m), $H_r$ is the riverbed elevation (m), $\partial h/\partial x$ is the slope of the groundwater level, $l$ is the connected length between catchments (m), $T$ is the average aquifer thickness (m), $A$ is the catchment area (m$^2$), $S$ is the storage coefficient, and $dh/dt$ is the rate of level change.

### 2.4. Model Calibration by Infiltration Methods and Comparison Criteria

The Boryeong Dam catchment was assumed to be a single catchment to calibrate and compare the model performances by three infiltration methods. Since the type of infiltration method applied to the model could significantly affect the results of the runoff simulation, it was necessary to simulate the runoff model for each infiltration method in advance and select the most suitable infiltration method for the study area. The CAT provides three infiltration analysis methods: the rainfall excess method, the Horton method, and the Green–Ampt method, with each infiltration method having unique parameters to be calibrated. Rainfall excess is that part of rainfall which is not lost to infiltration, interception, and depression storage. In the case of the rainfall excess infiltration method, it is assumed that infiltration occurs unconditionally until the groundwater zone is saturated

using unsaturated vertical and horizontal hydraulic conductivities, and the saturated soil moisture increases since it is assumed that the excess precipitation is surface runoff. The Green–Ampt infiltration method is calculated using the soil parameters, and the Horton infiltration method is calculated using the infiltration capacity curve equation as a function of time using empirical values. In the case of the rainfall excess method, the simulated runoff hydrograph tends to be the closest to the observed hydrograph, and the simulated peak flow tends to be the most underestimated among the three infiltration methods. In the case of the Green–Ampt method, the runoff ratio increases sharply according to the occurrence of rainfall, and it starts to decrease when the rainfall ends. The Green–Ampt infiltration method is based on the physical soil parameters; however, the simulated results tend to be highly sensitive to rainfall characteristics.

The SCEUA-P global optimization algorithm [35], which is a module of PEST (Model-Independent Parameter Estimation) and a parameter optimization and uncertainty analysis package [36] associated with the CAT model, was applied for parameter calibration while considering its applications, robustness, and efficiency, and the sensitive parameters for runoff simulation were selected as the calibration parameters.

In the rainfall excess infiltration method, the saturated soil moisture ($\theta s$), saturated vertical hydraulic conductivity ($ks$), and saturated horizontal hydraulic conductivity ($ksi$) were selected for parameter calibration. The saturated soil moisture ($\theta s$), saturated hydraulic conductivity ($ks$), saturated horizontal hydraulic conductivity ($ksi$), and wetting front soil suction head ($PSI$) were calibrated using the Green–Ampt infiltration method, and the maximum and minimum infiltration capacities ($fo$, $fc$) were calibrated using the Horton infiltration method. The saturated hydraulic conductivity of the riverbed ($ku\_riv$) and the storage coefficient of the aquifer ($aqf\_S$) were manually calibrated for each of the three infiltration methods (Table 1).

**Table 1.** Soil parameters for SCEUA-P global optimization, and river and aquifer parameters for manual calibration.

| Description | Parameter | Range | Unit |
|---|---|---|---|
| Soil | | | |
| Saturated soil moisture | s_per | 0.3–0.6 | - |
| Saturated vertical hydraulic conductivity | ks | 0.00001–0.08 | mm/s |
| Saturated lateral hydraulic conductivity | ksi | 0.00001–0.8 | mm/s |
| Wetting front soil suction head | PSI | 49.5–316.3 | mm |
| Maximum (initial) infiltration capacity | fo | 25–260 | mm/h |
| Minimum (asymptotic) infiltration capacity | fc | 0–12 | mm/h |
| River | | | |
| Saturated hydraulic conductivity of reverbed | ku_riv | 0.0001–0.000001 | mm/s |
| Aquifer | | | |
| Storage coefficient | S | 0.1–0.01 | - |

The model was calibrated for the period 2000–2009 and validated for the period 2010–2019. The Nash–Sutcliffe efficiency (NSE) index is commonly used as an indicator of hydrological model performance; however, its over-sensitivity to extreme values has been recognized [37,38]. The multi-objective functions used for evaluating model performances in this study were the Kling–Gupta efficiency ($KGE$), the determination coefficient ($R^2$), $NSE$, and the root mean square error ($RMSE$) [37]:

$$R^2 = 1 - \frac{\sum (Q_{obs} - Q_{sim})^2}{\sum (Q_{obs} - \overline{Q_{obs}})^2} \tag{7}$$

$$NSE = 1 - \frac{\sum (Q_{obs} - Q_{sim})^2}{\sum (Q_{obs} - Q_{mean})^2} \tag{8}$$

$$KGE = 1 - \sqrt{(R^2[Q_{obs}, Q_{sim}] - 1)^2 + \left(\frac{SD[Q_{sim}]}{SD[Q_{obs}]} - 1\right)^2 + \left(\frac{M[Q_{sim}]}{M[Q_{obs}]} - 1\right)^2} \qquad (9)$$

$$MSE = \sqrt{\frac{\sum(Q_{obs} - Q_{sim})^2}{n}} \qquad (10)$$

where $Q_{obs}$, $Q_{sim}$, and $Q_{mean}$ are the observed, simulated, and mean observed streamflow, respectively, and $SD$, $M$, and $n$ are the standard deviation, mean, and number of data points, respectively.

### 2.5. CAT Model Setup for 10 Sub-Catchments

In this study, the Boryeong Dam catchment was divided into five forest catchments and five paddy field catchments to analyze the groundwater hydrological response in each sub-catchment and the interrelationship between the paddy fields and the watershed. The watershed was divided into three regions (R1–R3) based on the daily groundwater pumping data [5]. The regions were then divided into 10 sub-catchments to analyze the groundwater level in detail. The watershed was assumed to consist of only paddy fields and forest areas. The sub-catchments were divided by the area ratio of paddy fields and forests of three regions according to the land use map [27]. The R1 (northwestern) region and the R3 (southern) region of the catchment were divided into one forest area and one paddy field area, while the R3 (northeastern) region of the watershed was divided into three forest catchments and three paddy catchments considering the amount of water intake from the agricultural reservoirs to the paddy fields (Table 2).

**Table 2.** Areas and slopes of each sub-catchment in the Boryeong Dam catchment.

| Regions | R1 | | R2 | | | | | | R3 | |
|---|---|---|---|---|---|---|---|---|---|---|
| Sub-Catchments | F 1 | P 1 | F 2 | F 2-1 | F 2-2 | P 2 | P 2-1 | P 2-2 | F 3 | P 3 |
| Area (km$^2$) | 40.1 | 0.9 | 44.8 | 1.8 | 2.1 | 5.8 | 0.8 | 0.8 | 61.2 | 4.8 |
| Slope (-) | 0.26 | 0.15 | 0.40 | 0.30 | 0.28 | 0.19 | 0.16 | 0.15 | 0.23 | 0.11 |

F: forest, P: paddy.

In addition, we considered additional potential factors that could affect the hydrological cycle of the watershed. As shown in Figure 4, we considered the water export to the Boryeong downtown through the Gaehwa weir, which is located outside the catchment in the northwestern region of the catchment. The Samsan and Hwasung agricultural reservoirs in the northeastern region were considered as the water supplied from these reservoirs directly affects the paddy fields in the catchment. In addition, a water supply from the Geum River into the watershed was also considered.

Groundwater movement connections between adjacent sub-catchments were considered by linking the upslope forest catchment and downslope paddy field catchment (Figure 5). The groundwater movement connection parameters were the aquifer slope (%), the average distance between sub-catchments (m), the length of the boundary line (m), and the hydraulic conductivity of the aquifer (mm/s). The F 1 catchment was connected to the P 1 catchment, the F 2, F 2-1, and F 2-2 catchments were connected to the P 1, P 2, and P 2-2 catchments, respectively, and the F 3 catchment was connected to the P 3 catchment.

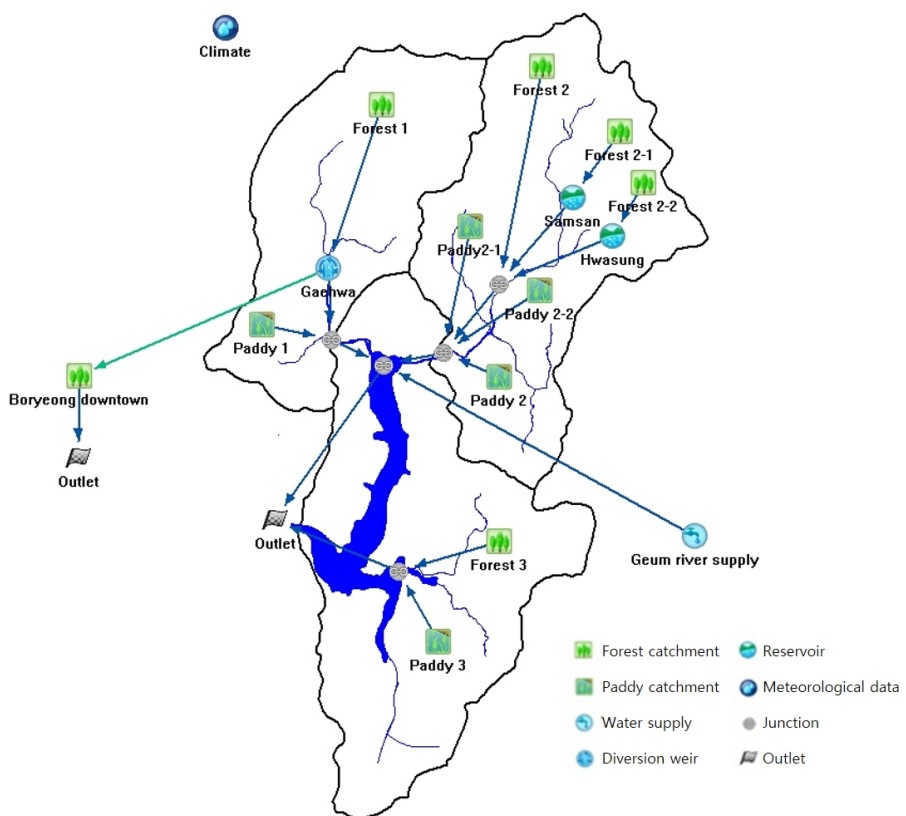

**Figure 4.** CAT model setup of the Boryeong Dam catchment including 10 sub-catchments and water export and import.

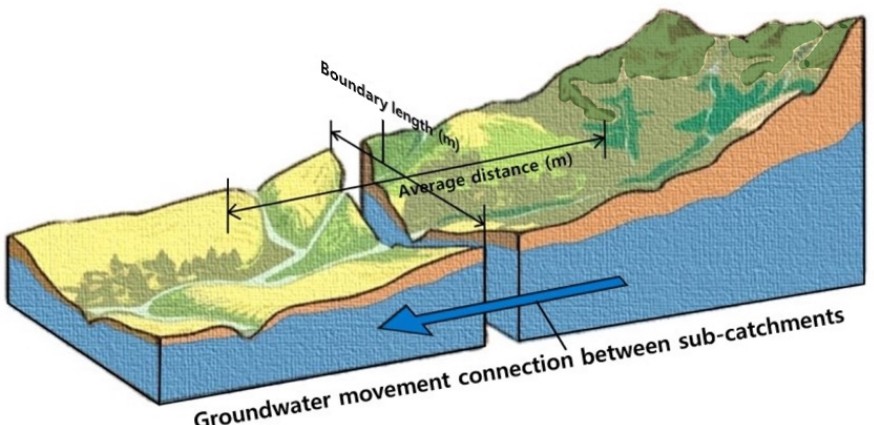

**Figure 5.** Groundwater movement connections between paddy field and forest catchments.

*2.6. Human Impact on Hydrological Cycle in Boryeong Dam Catchment*

2.6.1. Water Supply Tunnel from Geum River

Due to the severe drought that occurred in 2015, the middle-western part of Korea was severely affected by water shortages. The water storage rate of the Boryeong Dam continued to decrease despite the implementation of restrictions on water use in the catchment. The average inflows of the Boryeong Dam in 2014 and 2015 were 2.32 m$^3$/s and 1.93 m$^3$/s, respectively, and remained at about 50% of the average inflow (4.33 m$^3$/s) for 18 years from 1998 to 2015. In response, the Korean government constructed an emergency water supply tunnel facility that connects the lower reaches of the Geum River and the upper Boryeong Dam, which has been operational since February 2016. The length of the water supply tunnel is approximately 21 km, and it can supply up to 115,000 m$^3$/day of

water to the Boryeong Dam (Figure 6). The Geum River water supply was calculated using the daily inflow time series data provided by the Korea Water Resources Corporation [5] using the module for external water supply in the CAT.

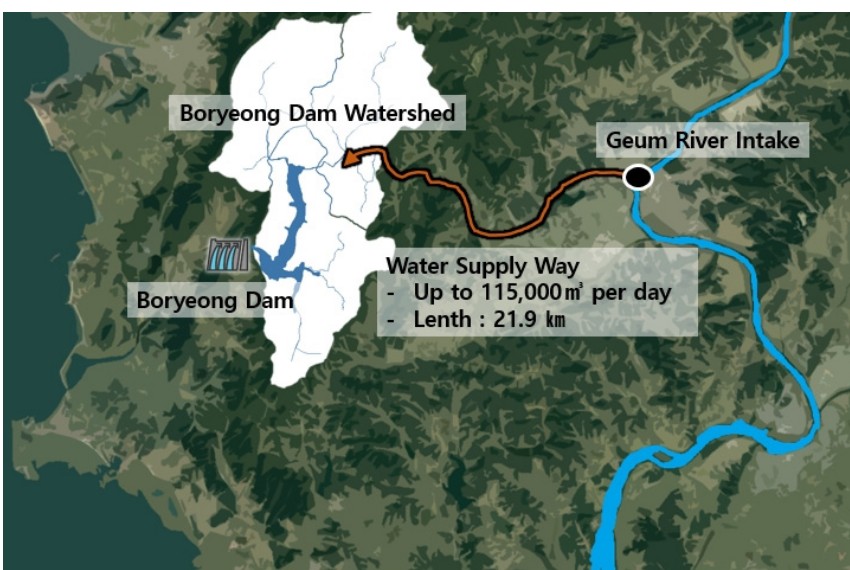

**Figure 6.** Water supply tunnel from the lower reaches of the Geum River to the upper Boryeong Dam catchment.

### 2.6.2. Groundwater Pumping Rate

The groundwater use data for the 10-year period from 2008 to 2017 were provided by the National Groundwater Information Center [16], and the average daily groundwater use data were applied to the study period. The daily groundwater use for agricultural, industrial, and domestic water use was classified, and the volume of groundwater for each purpose was divided by the area ratio in each sub-catchment. Agricultural groundwater was applied to the paddy fields, and domestic and industrial groundwater use was applied to the forest areas. In the case of the R1 region, the agricultural groundwater use rate was 96% of the total groundwater consumption over 2000 to 2019, and the rates were 93.8% and 44.3% for the R2 and R3 regions, respectively. Table 3 shows the daily groundwater use per sub-catchment according to the area ratio. The average daily volume of groundwater pumping was applied as a constant value in the model. Groundwater pumping is only applied in the period from April to September since irrigation begins in early April and continues until the end of September in Korea.

**Table 3.** Daily groundwater use distributed according to the area ratio of sub-catchments in the irrigation period.

| Sub-Catchment | F 1 | F 2 | F 2-1 | F 2-2 | F 3 |
|---|---|---|---|---|---|
| Groundwater use (m³/day) | 1328 | 491 | 20 | 22 | 952 |
| | **P 1** | **P 2** | **P 2-1** | **P 2-2** | **P 3** |
| Groundwater use (m³/day) | 1055 | 10,065 | 1388 | 1392 | 14,553 |

### 2.6.3. Reservoirs in the Catchment

In the Boryeong Dam catchment, there is a Gaewha diversion weir structure in the northwestern area of the catchment, which supplies water to the Boryeong downtown, located out of the catchment. There are two reservoirs upstream of the northeastern area of the catchment that are used to supply agricultural water to nearby paddy fields: the Samsan reservoir supplies agricultural water to P 2-1, and the Hwasung reservoir supplies water to P 2-2. The intake volume of the Gaewha diversion weir is 5240 m³/day, and this

was applied as a constant value. The agricultural water supply data from the Samsan and Hwasung reservoirs were collected as a time series from 2012 to 2018, and the intake water supply for the paddy fields was considered only during the cultivation period from April 1 to September 31. Table 4 shows the average daily intake rate of the reservoirs.

**Table 4.** Amount of average daily intake rate for the Samsan and Hwasung agricultural reservoirs during the cultivation period.

| Intake | 2012 | 2013 | 2014 | 2015 | 2016 | 2017 | 2018 |
|---|---|---|---|---|---|---|---|
| Samsan (m$^3$/day) | 5118 | 4431 | 4257 | 3041 | 1841 | 3744 | 3394 |
| Hwasung (m$^3$/day) | 5165 | 4308 | 4916 | 4771 | 3477 | 2263 | 3151 |

*2.7. Wet and Dry Periods of the Catchment*

To compare and analyze the hydrological responses and groundwater level changes in the wet period and the drought period in the catchment, consecutive wet and dry years were selected based on the average annual precipitation and the average annual runoff ratio within the validated period of 2010–2019. The 30-year average annual precipitation in the Boryeong Dam catchment is approximately 1244 mm, and the 20-year average annual runoff ratio is approximately 53.4%. Three consecutive years of 2010, 2011, and 2012 were selected as the wet period, with annual precipitation rates of 1467 mm, 1971 mm, and 1588 mm, respectively, and with runoff ratios of 64.8%, 75.3%, and 54.3 %, respectively, indicating higher than average precipitation and runoff ratios. The three consecutive years of 2014, 2015, and 2016 were selected as the dry period, with annual precipitation rates of 1089 mm, 1023 mm, and 1086 mm, respectively, and with runoff ratios of 40.4%, 36.2%, and 47.4%, respectively, indicating lower than average precipitation and runoff ratios (Figure 7).

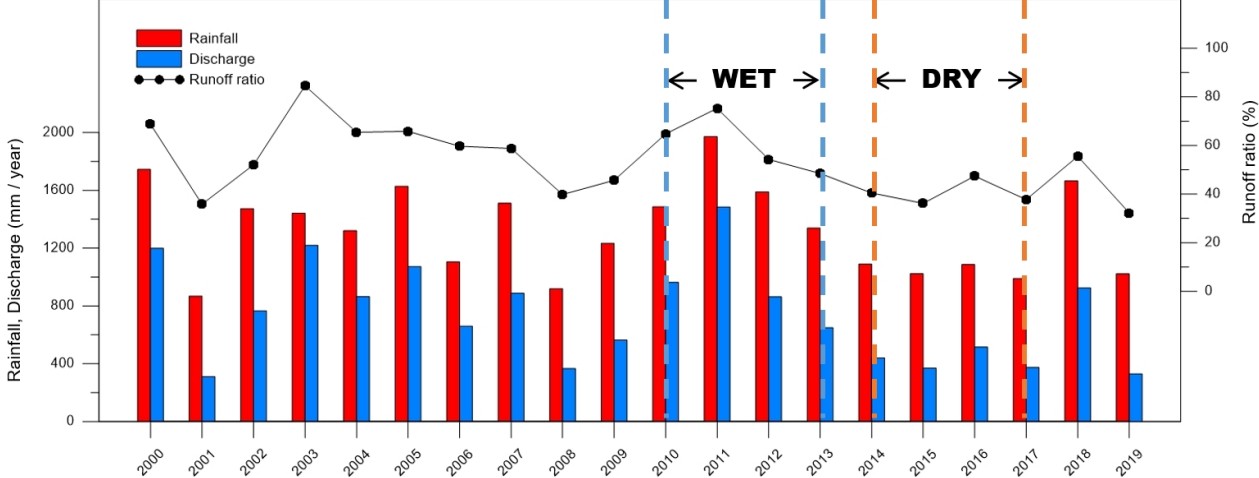

**Figure 7.** Comparison of annual precipitation, discharge, and runoff ratio in the wet and dry periods.

## 3. Results

*3.1. Model Calibration for Single Watershed Simulation Using Three Different Infiltration Methods*

The parameter values before and after calibration for the rainfall excess method, Green–Ampt method, and Horton infiltration method are shown in Table 5. The $\theta s$, $ks$, and *ksi* parameters were calibrated using the rainfall excess method. The $\theta s$, $ks$, *ksi*, and *PSI* parameters were calibrated using the Green–Ampt method, and the *fo* and *fc* parameters were calibrated using the Horton method.

**Table 5.** Parameter values before and after calibration for the rainfall excess method, Green–Ampt method, and Horton infiltration method.

| | Rainfall Excess | | | Green–Ampt | | | | Horton | |
| --- | --- | --- | --- | --- | --- | --- | --- | --- | --- |
| | $\theta s$ | $ks$ | $ksi$ | $\theta s$ | $ks$ | $ksi$ | $PSI$ | $fo$ | $fc$ |
| Default | 0.3748 | 0.0006 | 0.002 | 0.3748 | 0.0006 | 0.002 | 150 | 5 | 150 |
| Calibrated | 0.4461 | 0.004 | 0.001 | 0.4988 | 0.0239 | 0.0423 | 248.64 | 9.88 | 199.62 |

Table 6 shows the statistical results for the observed and simulated discharge before and after parameter optimization for each infiltration analysis method, assuming the Boryeong Dam catchment as a single catchment. In all cases, before and after parameter calibration, the NSE results, when applying the rainfall excess infiltration analysis method, show relatively high accuracy. After parameter calibration, the NSE results are 0.802 for rainfall excess, 0.625 for Green–Ampt, and 0.762 for the Horton infiltration analysis method, and the results for KGE are 0.827, 0.775, and 0.832, respectively. In the case of KGE, the results of the Horton infiltration method are similar or show slightly higher accuracy than the results of rainfall excess; however, the model performance when applying the rainfall excess infiltration method was considered to be satisfactory considering the $R^2$ and RMSE results together. Therefore, we assumed the rainfall excess infiltration method to be the most suitable infiltration method for analyzing the groundwater level of the watershed in the detailed sub-catchment model setup.

**Table 6.** Comparison of statistical results against observed and simulated streamflow using three infiltration analysis methods.

| Rainfall Excess | $R^2$ | NSE | KGE | RMSE |
| --- | --- | --- | --- | --- |
| Calibration | 0.803 | 0.802 | 0.827 | 7.046 |
| Validation | 0.781 | 0.722 | 0.825 | 6.667 |
| **Green–Ampt** | $R^2$ | NSE | KGE | RMSE |
| Calibration | 0.711 | 0.625 | 0.775 | 9.691 |
| Validation | 0.722 | 0.475 | 0.616 | 9.163 |
| **Horton** | $R^2$ | NSE | KGE | RMSE |
| Calibration | 0.768 | 0.762 | 0.832 | 7.719 |
| Validation | 0.784 | 0.720 | 0.813 | 6.688 |

A comparison of the observed and simulated streamflow for the calibrated and validated periods on a logarithmic scale is shown in Figure 8a. The $R^2$, NSE, and KGE values were 0.803, 0.802, and 0.827, respectively, during the calibration period of 2000–2009, and the values of $R^2$, NSE, and KGE were 0.781, 0.722, and 0.825, respectively, during the validation period of 2010–2019 (Figure 8b). The statistical results of the calibration period show a slightly higher accuracy for the model simulation compared to the results of the validation period; however, both the calibration and validation periods had satisfactory model performances.

*3.2. Comparison of Streamflow in Wet and Dry Periods*

We simulated the 10 sub-catchment models and compared the streamflow results against the observed streamflow at the outlet of the watershed. The annual comparisons of the observed and simulated streamflow at the outlet of the watershed in each year of the wet period are shown in Figure 9a, and those of the dry period are shown in Figure 9b. In the case of the wet period, the annual observed and simulated streamflows were 1491 m³/s and 1825 m³/s in 2010, 2435 m³/s and 2812 m³/s in 2011, and 1597 m³/s and 1634 m³/s in 2012, respectively. In the dry period, the annual observed and simulated streamflows were 754 m³/s and 834 m³/s in 2014, 617 m³/s and 701 m³/s in 2015, and 835 m³/s and 976 m³/s in 2016, respectively. The simulated streamflow discharge tended to be

overestimated in the wet period compared to the observed streamflow; however, it was slightly underestimated during the dry period.

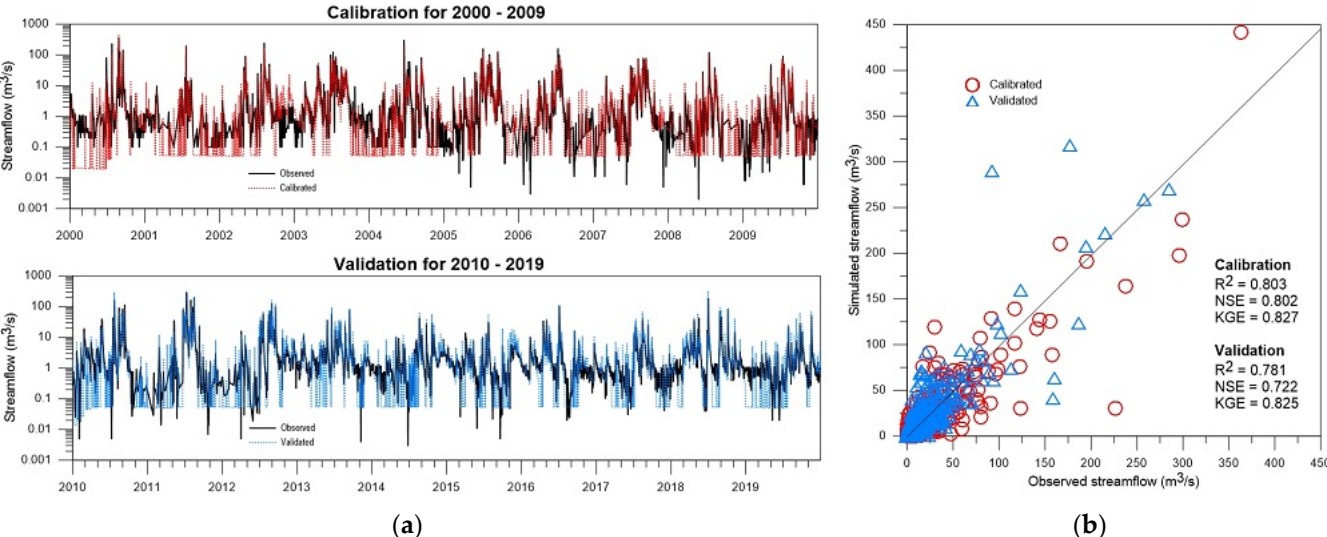

**Figure 8.** (**a**) Comparison of observed, calibrated (2000–2009), and validated (2010–2019) streamflow of the rainfall excess infiltration method-applied model. (**b**) Scattered plot of calibration and validation against observed and simulated streamflow.

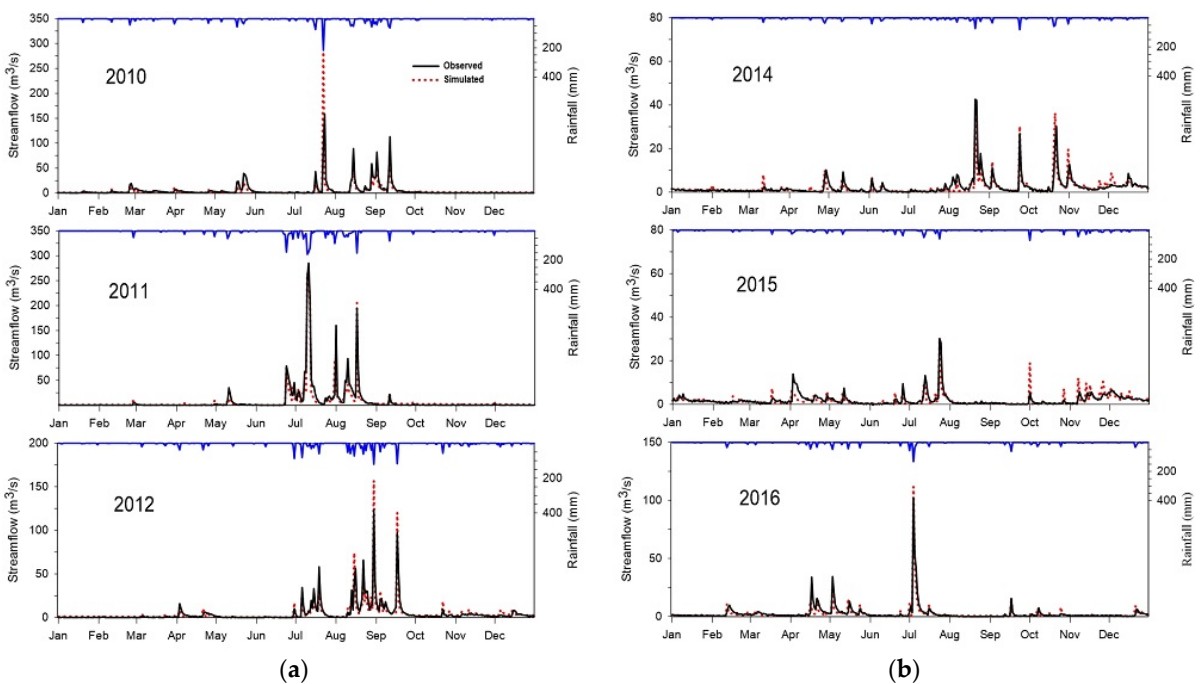

**Figure 9.** (**a**) The comparison of observed and simulated streamflow and precipitation for each year in the wet period. (**b**) The comparison of observed and simulated streamflow and precipitation for each year in the dry period.

The comparisons of the observed and simulated streamflow for the entire wet period and the entire dry period are shown as a logarithmic scale in Figure 10a,b, respectively. The statistical results for the wet period show values of $R^2$, NSE, KGE, and RMSE of 0.74, 0.73, 0.80, and 10.25, respectively, and the statistical results for the dry period show values of $R^2$, NSE, KGE, and RMSE of 0.76, 0.76, 0.80, and 2.54, respectively. The results of both the wet and dry period simulations show satisfactory model performances, and the simulation of the dry period showed a slightly better model performance compared to the wet period.

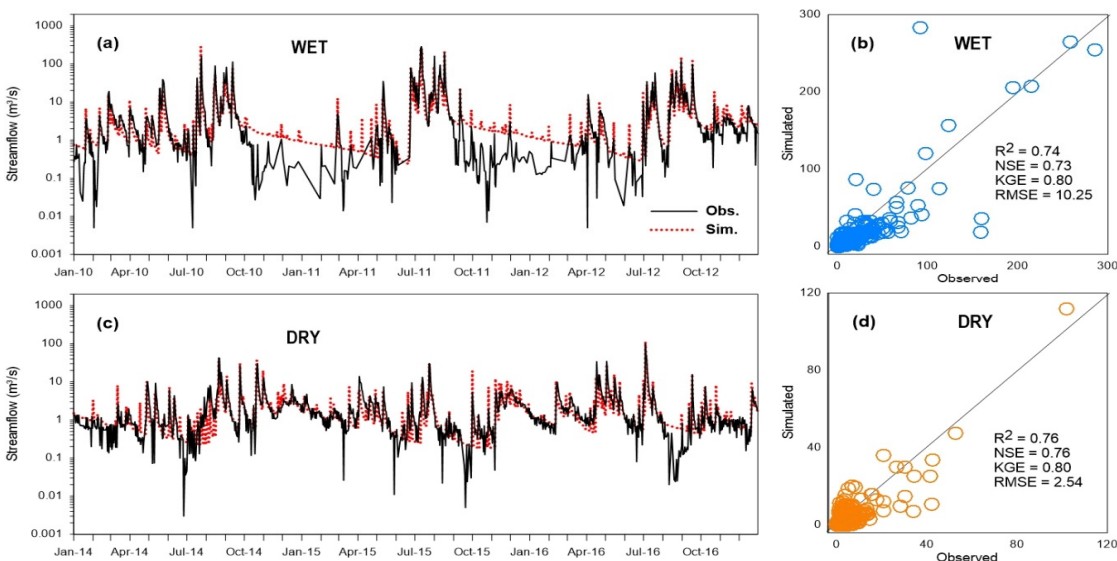

**Figure 10.** (**a**) The observed and simulated streamflow plotted on a logarithmic scale for the wet period. (**b**) The statistical model performance of the wet period. (**c**) The observed and simulated streamflow plotted on a logarithmic scale for the dry period. (**d**) The statistical model performance of the dry period.

### 3.3. Groundwater Level of Wet and Dry Periods

To analyze the differences in the groundwater level between the drought and wet periods, it is necessary to closely compare the monthly groundwater level fluctuations in each period. Table 7 shows the difference between the monthly average groundwater level for the three years of the wet period and the dry period in forest sub-catchments, and Table 8 shows the difference between the monthly average groundwater levels for the three years of the wet and dry periods in the paddy sub-catchments. The differences indicate that the groundwater level decreases in the forest sub-catchments were larger than the groundwater level decreases in the paddy sub-catchments. In the case of the forest sub-catchments, the decrease in the annual average groundwater level for each sub-catchment shows that the minimum value was 0.01 m in the F 2-1 catchment, and the maximum value was 0.38 m in the F 3 catchment. In the case of the paddy sub-catchments, the minimum value was 0.09 m in the P 2 and P 2-1 catchments, and the maximum value was 0.12 m in the P 3 catchment. The monthly groundwater level decrease in the dry period compared to the wet period in the forest sub-catchments shows that the maximum difference was 1.07 m in the F 3 catchment in September, and the monthly groundwater level decrease in the paddy sub-catchments shows that the maximum difference was 0.37 m in the P 3 catchment in September, indicating that the maximum annual decrease in the groundwater level in the forest sub-catchments was about 2.9 times greater than that of the paddy sub-catchments.

The difference in the groundwater level of the paddy sub-catchments in the comparison of the wet and dry periods was less than the difference in the groundwater level of the forest sub-catchments due to the reduced agricultural groundwater use during the dry period (Table 9). In the case of P 1, there were no significant differences in the use of agricultural groundwater between the wet and dry periods as the area of the catchment is smaller than that of the other paddy sub-catchments. However, in the case of the paddy sub-catchments P 2, P 2-1, P 2-2, and P 3, groundwater use was greater during the wet period since the paddy sub-catchment areas are larger, indicating that agricultural groundwater use sharply decreased during the dry period (Table 9). In the case of P 2, the annual average groundwater use during the wet and dry periods was 14,773 m$^3$/day and 3164 m$^3$/day, respectively, indicating that only 21.4% of the groundwater was used in the dry period compared to the wet period. In the case of P 2-1 and P 2-2, groundwater use in the dry period was approximately 21.4% of that of the wet period. In the case of P 3, the

annual average groundwater use during the wet and dry periods was 22,636 m³/day and 2634 m³/day, respectively, indicating that only 11.6% of the groundwater was used in the dry period compared to the wet period. The results show that the available groundwater for agriculture is greatly affected by the drought season, and the efficient management of groundwater is very important, especially in groundwater-fed irrigation catchments.

**Table 7.** The differences in monthly average groundwater elevation between the wet and dry periods in the forest sub-catchments (m).

| Month | F 1 | F 2 | F 2-1 | F 2-2 | F 3 |
|---|---|---|---|---|---|
| Jan | 0.03 | −0.06 | −0.38 | −0.03 | 0.06 |
| Feb | 0.01 | −0.06 | −0.41 | −0.07 | 0.04 |
| Mar | 0.02 | −0.04 | −0.42 | −0.08 | 0.04 |
| Apr | 0.02 | −0.02 | −0.41 | −0.07 | 0.03 |
| May | −0.03 | −0.06 | −0.49 | −0.15 | −0.02 |
| Jun | 0.01 | −0.01 | −0.45 | −0.12 | 0.02 |
| Jul | 0.27 | 0.22 | −0.19 | 0.14 | 0.29 |
| Aug | 0.65 | 0.53 | 0.26 | 0.55 | 0.68 |
| Sep | 1.01 | 0.80 | 0.74 | 0.94 | 1.07 |
| Oct | 0.90 | 0.61 | 0.75 | 0.85 | 0.99 |
| Nov | 0.67 | 0.38 | 0.63 | 0.67 | 0.78 |
| Dec | 0.47 | 0.20 | 0.47 | 0.49 | 0.58 |
| Average | 0.34 | 0.21 | 0.01 | 0.26 | 0.38 |

**Table 8.** The differences in monthly average groundwater elevation between the wet and dry periods in the paddy sub-catchments (m).

| Month | P 1 | P 2 | P 2-1 | P 2-2 | P 3 |
|---|---|---|---|---|---|
| Jan | 0.04 | 0.06 | 0.05 | 0.06 | 0.10 |
| Feb | 0.02 | 0.03 | 0.02 | 0.03 | 0.07 |
| Mar | 0.00 | 0.01 | −0.01 | 0.00 | 0.02 |
| Apr | −0.01 | −0.01 | −0.02 | −0.01 | 0.00 |
| May | −0.03 | −0.03 | −0.05 | −0.04 | −0.04 |
| Jun | −0.03 | −0.03 | −0.05 | −0.03 | −0.04 |
| Jul | 0.01 | 0.01 | 0.00 | 0.01 | 0.03 |
| Aug | 0.15 | 0.13 | 0.14 | 0.16 | 0.18 |
| Sep | 0.32 | 0.27 | 0.31 | 0.33 | 0.37 |
| Oct | 0.29 | 0.25 | 0.30 | 0.31 | 0.35 |
| Nov | 0.22 | 0.19 | 0.23 | 0.23 | 0.26 |
| Dec | 0.17 | 0.15 | 0.18 | 0.18 | 0.21 |
| Average | 0.10 | 0.09 | 0.09 | 0.10 | 0.12 |

**Table 9.** Groundwater pumping rate for the wet and dry periods by sub-catchment (m³/day).

|  | Wet | | | | Dry | | | |
|---|---|---|---|---|---|---|---|---|
|  | 2010 | 2011 | 2012 | Avg. | 2014 | 2015 | 2016 | Avg. |
| F 1 | 873 | 1710 | 1703 | 1429 | 1449 | 1474 | 1487 | 1470 |
| F 2 | 371 | 396 | 501 | 423 | 503 | 537 | 796 | 612 |
| F 2-1 | 15 | 16 | 20 | 17 | 20 | 22 | 32 | 25 |
| F 2-2 | 17 | 18 | 23 | 19 | 23 | 25 | 36 | 28 |
| F 3 | 613 | 933 | 941 | 829 | 1259 | 1317 | 1333 | 1303 |
| P 1 | 559 | 1377 | 1377 | 1104 | 1154 | 1166 | 1199 | 1173 |
| P 2 | 13,511 | 14,781 | 16,028 | 14,773 | 3286 | 3323 | 2882 | 3164 |
| P 2-1 | 1864 | 2039 | 2211 | 2038 | 453 | 458 | 398 | 436 |
| P 2-2 | 1868 | 2044 | 2216 | 2043 | 454 | 459 | 399 | 437 |
| P 3 | 22,320 | 22,778 | 22,810 | 22,636 | 2480 | 2698 | 2725 | 2634 |

In Korea, approximately 30% of the annual precipitation is concentrated over a month, from late June to late July, during the rainy season [25], and irrigation begins in early April and continues until the end of September. Therefore, to analyze the fluctuations in the groundwater level, it is necessary to closely examine the fluctuations in the groundwater level during the rainy seasons but also the irrigation period when agricultural groundwater is used. The monthly groundwater level fluctuations during the three years of the wet and dry periods are shown in Figure 11 for the forest sub-catchments and in Figure 12 for the paddy sub-catchments. In the case of the forest sub-catchments in the wet period, the groundwater level gradually decreased from January to June, then started to rise from the end of June—the rainy season in Korea—and reached the highest groundwater level in July. During the dry period in forest sub-catchments, the resulting graphs show a marked difference, showing a lower groundwater level than that of the wet period; however, there was no significant difference in the monthly groundwater level because of the lack of baseflow due to insufficient precipitation in the rainy season. In the case of the paddy catchments in the wet period, the groundwater level started to decrease from April, since the irrigation period is from April to September in Korea. The groundwater level then recovered rapidly, rising sharply in August in response to the precipitation of the rainy season starting from the end of June. The changes in the groundwater level in response to precipitation were slower and had a longer time lag than those of the forest catchments due to the agricultural groundwater pumping in the paddy catchments. During the dry period in the paddy catchments, the groundwater level started to decrease from April and could not recover quickly in July, as it did in the wet period, due to insufficient precipitation during the rainy season. The groundwater level gradually recovered from September after the irrigation period ended.

## 4. Conclusions and Summary

The impact of drought on groundwater levels over the Boryeong Dam catchment was analyzed in the present study. We conducted a single watershed runoff simulation for 2000–2019 using three infiltration methods to calibrate the parameters and determine the most suitable infiltration method for the study area. Runoff simulations with three infiltration methods were calibrated for 2000–2009 and validated for 2010–2019 against the observed streamflow. The results of the single watershed runoff simulation show that the rainfall excess infiltration method showed the most satisfactory model performance compared to the Horton and Green–Ampt infiltration methods, with values of 0.802 for NSE and 0.827 for KGE in the calibrated period, and 0.722 for NSE and 0.825 for KGE in the validated period. Then, the Boryeong Dam catchment model with the rainfall excess infiltration method was divided into 10 sub-catchments to analyze the hydrological cycle processes in paddy field and forest sub-catchments and the groundwater level changes in drought periods. The wet period of 2010–2012 and the dry period of 2014–2016 were selected based on the average annual precipitation and the average annual runoff ratio. In addition, the model considered the human impacts on the hydrological cycle over the study area, such as reservoirs, the agricultural groundwater pumping rate, and the import or export of water in the catchment.

The annual comparisons of the observed and simulated streamflow for each year showed that streamflow was overestimated by the model in the wet period and slightly underestimated in the dry period. In the case of the wet period, the statistical results of $R^2$, NSE, KGE, and RMSE were 0.74, 0.73, 0.80, and 10.25, respectively, while in the case of the dry period, these values were 0.76, 0.76, 0.80, and 2.54, respectively, indicating that both model performances for the wet and dry periods showed a satisfactory model performance.

The differences in the groundwater level in the forest catchments were larger than those in the paddy field catchments, as groundwater use in the dry period in the paddy sub-catchments was 11.6% compared to 21.4% in the wet period.

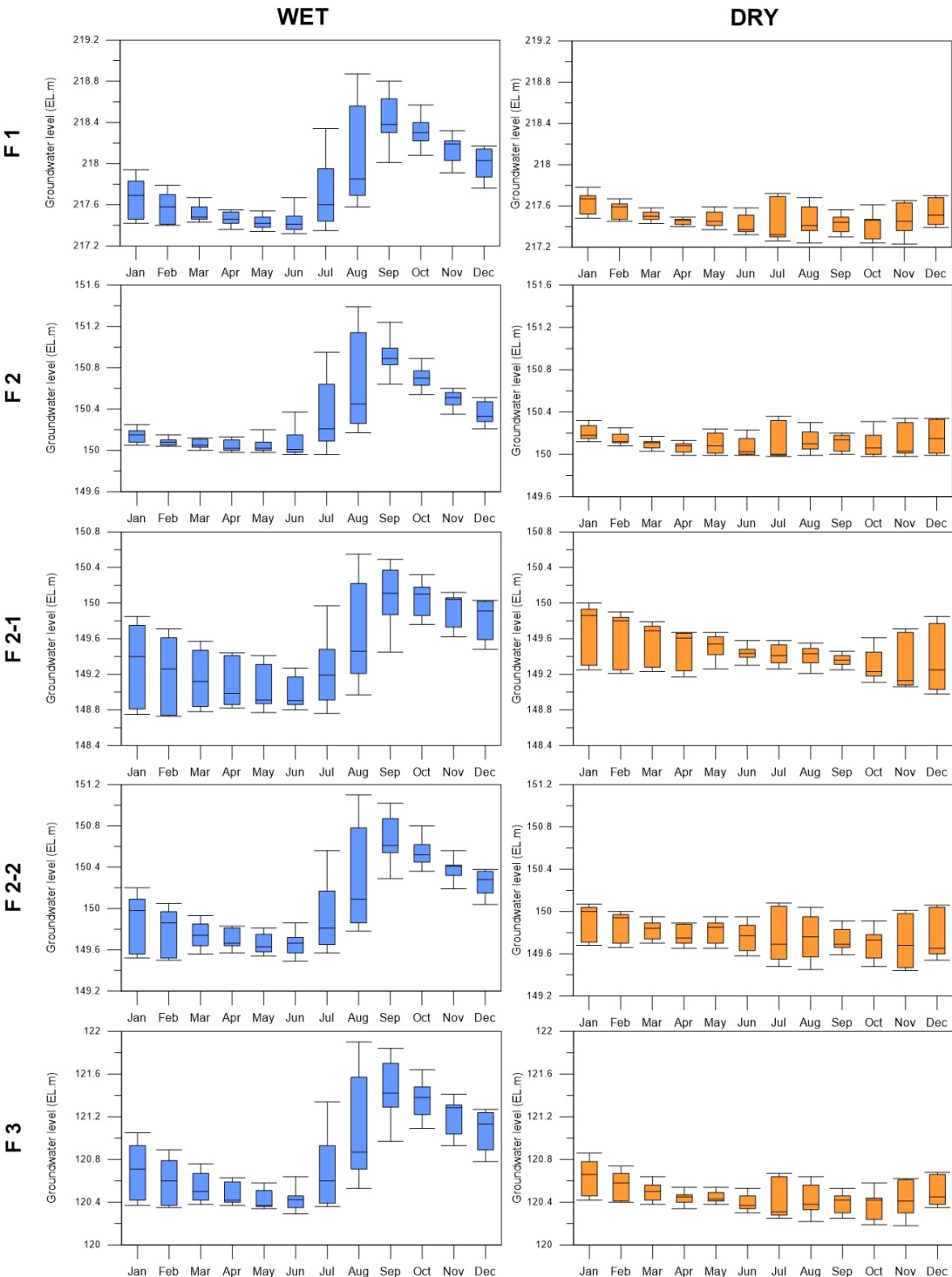

**Figure 11.** The monthly average groundwater level changes in the forest sub-catchments in the wet and dry periods.

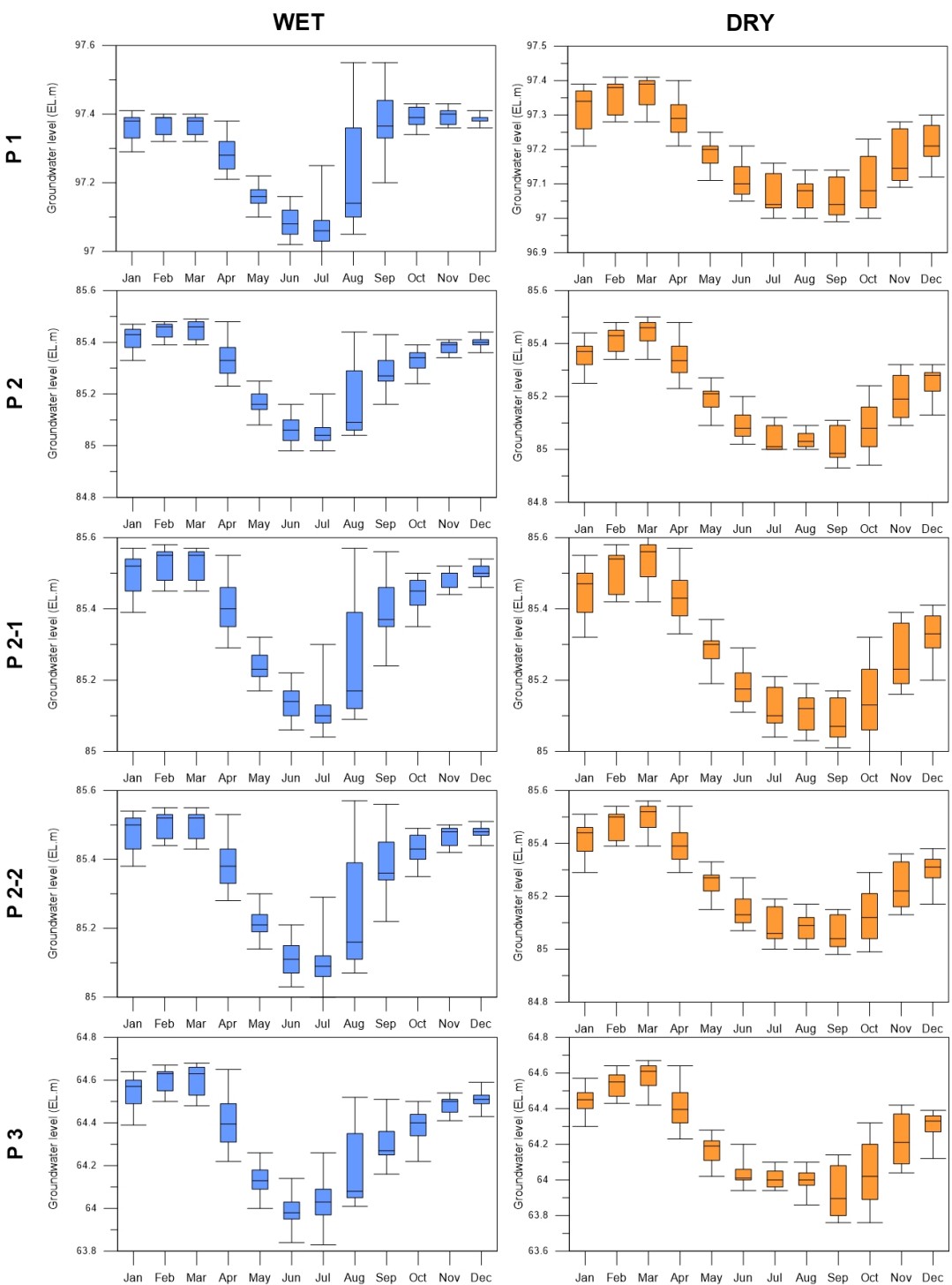

**Figure 12.** The monthly average groundwater level changes in the paddy sub-catchments in the wet and dry periods.

During the wet period in the forest sub-catchments, the groundwater level continued to decrease from January to May and started to increase from June—the rainy season—showing the highest groundwater level in July. During the dry period, the groundwater levels did not significantly change in the forest catchments. In the case of the paddy field catchments, the groundwater level affected by the irrigation period recovered rapidly, showing a sharp increase in August in response to the rainy season during the wet period; however, it recovered slowly during the dry period due to the lack of precipitation in the rainy season. The agricultural groundwater use was limited during the drought period to

11.6–21.4% compared to the normal period, and the recovery of the groundwater level in the paddy field catchments took a longer time.

Groundwater is the main source of fresh water required for socio-economic development in the region for both domestic and agricultural use. The sustainable use of agricultural groundwater depends on understanding the dynamic behavior and responses of groundwater in paddy catchments and their relationship to other hydrological components. Since groundwater and surface water flows have strong interrelationships, they need to be considered and managed in combination. The use of integrated surface–groundwater modeling analysis is necessary to model groundwater availability under present and future climate conditions and to ensure the sustainability of groundwater use.

The results of this study reveal that agricultural groundwater availability is dependent on the prevailing climatic conditions. It is necessary to secure sustainable groundwater resources through efficient water management at the catchment scale and to compare the hydrological cycle processes between the mountainous catchments and paddy field catchments in Korea. Subsequent studies are needed to accurately analyze the hydrological cycle process of paddy field catchments by analyzing the relationship between precipitation, groundwater levels, baseflow, and groundwater recharge.

**Author Contributions:** Conceptualization, S.P., H.K.; Methodology, S.P., H.K. and C.J.; Formal analysis, S.P.; Investigation, C.J.; Resources, S.P., C.J.; Data curation, S.P., H.K.; Writing—original draft preparation, S.P.; Writing—review and editing, H.K.; Supervision, H.K., C.J.; Project administration, H.K.; Funding acquisition, H.K. All authors have read and agreed to the published version of the manuscript.

**Funding:** This research was funded by the Korea Institute of Civil Engineering and Building Technology, grant number 20210194-001.

**Institutional Review Board Statement:** Not applicable.

**Informed Consent Statement:** Not applicable.

**Data Availability Statement:** This study did not include any publicly available datasets.

**Conflicts of Interest:** The authors declare no conflict of interest.

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
