# Peer review of "Impact of Groundwater Abstraction on Hydrological Responses during Extreme Drought Periods in the Boryeong Dam Catchment, Korea"

_water, doi:10.3390/w13152132_

Round 1

Reviewer 1 Report

Comments on Impact of Groundwater Abstraction on Hydrological Responses during Extreme Drought Periods in the Boryeong Dam Catchment, Korea:

  • Add some of the most important quantitative results to the Abstract.
  • Add/Replace the name of the study area to the Keywords.
  • In the last paragraph of the Introduction, the authors should clearly mention the weakness point of former works (identification of the gaps) and describe the novelties of the current investigation to justify us the paper deserves to be published in this journal.
  • In the first sentence of the Introduction, cite this recent useful paper on the importance of climate change and global warming to improve the literature and to show the importance of your work:

Global surface temperature: A new insight

  • Discuss the main reasons for the variations of the streamflow of Rainfall Excess infiltration method.
  • Focus on the advantages/disadvantages of the proposed method with respect to the obtained results.
  • How can expand the results in other regions with similar/different climates?
  • The quality of the language needs to improve by a native English speaker for grammatically style and word use.

Author Response

Dear reviewer,

We really appreciate for your review and guidance on our paper work.

I hope the language quality was improved by MDPI english editing service.

I also revised the paper according to your report, and kindly ask you to see the attachment.

Every detail answers can be found at the last page of the attached file and also can be found in the paper (red letters).

Thank you very much.

Best regards,

Sanghyun Park

Reviewer 2 Report

An interesting paper investigating the connections between surface and groundwater under drought conditions, in a catchment with high levels of agricultural groundwater use. This is an important area of study, with direct implications for water management now and under future climate and socio-economic changes.
Overall, the paper is well written: clear and logical, with well-presented results which support the conclusions. I have a number of small comments throughout, with the Materials and Methods section requiring most work.

Introduction:
Line 33: better to say: "in the preceding X months" to describe the period with no precipitation
Line 61: replace "also" with "can also be", since this statement can be true but is not universally true
Line 65-66: not entirely sure what this sentence is trying to say. Would also recommend moving this to be the second sentence in the paragraph, so that there's a distinction between general and geographically specific statements 

Materials and Methods:
Section 2.1: there are no details on the streamflow data - specifically the gauge location!
Line 137, 138, add some commas: conceptual, parameter-based, lumped hydrological model
physical, parameter-based, distributed hydrological model
Line 137: either add a couple of references where the previous model has been used, or remove the first sentence. 
Line 167-169: units do not match up in equation 1 (drainage coefficient must be in sqrt(m)/s) 
Equations 3 & 4: change Qr so that it doesn't represent two things.
Equation 6: dh/dt not defined below
Section 2.4: need to be made clear here that a single catchment run was performed to calibrate the infiltration methods (rather than the 10 sub-catchments mentioned in line 185). 
Should also justify why a single catchment run was sufficient to chose the correct infiltration method. 
Line 191-194: confusing, please rephrase
Line 204: reference 35 seems to be out of place
Line 225: change "number" to "number of data points"
Section 2.5: explain how climate is downscaled for 10 subcatchments
Line 231: include the reference for groundwater pumping data here
Section 2.6.2: apportioning domestic and industrial groundwater use by area in forest subcatchments, and only abstracting between April and September, are approximations that should be justified.
Section 2.6.3: the Gaewha diversion weir, and two agricultural reservoirs, are described along with their implementation in the model. However, assuming that the observed streamflow in figures 8- 10 is at the Boryeong dam outlet, how is the impact of this dam accounted for in the model?

Results:
Table 6: Before calibration values are not pertinent, I would leave them out
Section 3.2: Specify that section 3.2 is discussing results from main run (10 subcatchment) rather than calibration runs

Conclusion and Summary: clear, supported by the data.

References:
Reference 18 is indented

Figures:
There are two "figure 6", and several figures are not referenced in the text
Figure 3, what are Is and Ig?
Figure 4 is unhelpful (parameters not explained in caption, "Geum Import" not explained in caption, ksi only listed for rainfall excess but in the text is listed for Green-Ampt) 
Figure 7: (mm) to (mm/year) in y-axis label

Author Response

(The authors gave the same response as above.)

Round 2

Reviewer 1 Report

The authors have addressed to the comments. The quality of the manuscript is acceptable now. Congrats!

Author Response

Dear reviewer,

I really appreciate your comments!

Thank you very much.

Best regards,

Sanghyun Park